# Concurrent Training Programming: The Acute Effects of Sprint Interval Exercise on the Subsequent Strength Training

**DOI:** 10.3390/sports10050075

**Published:** 2022-05-10

**Authors:** Bertrand Mathieu, Julien Robineau, Julien Piscione, Nicolas Babault

**Affiliations:** 1French Rugby Union Federation, 3-5 rue Jean de Montaigu, 91463 Marcoussis, France; bertrand.mathieu@ffr.fr (B.M.); julien.robineau@ffr.fr (J.R.); julien.piscione@ffr.fr (J.P.); 2Center for Performance Expertise, CAPS, U1093 INSERM, Faculty of Sport Sciences, University of Bourgogne-Franche-Comté, 3 Allée des Stades Universitaires, BP 27877, 21078 Dijon, France

**Keywords:** cycling, running, fatigue, rugby union, strength, aerobic

## Abstract

Exercise modality has been proposed to reduce the interferences between aerobic and resistance sessions during concurrent training. The aim of the study was to examine the acute effects of cycling or running sprint interval exercise on subsequent resistance training sessions. Twenty-five competitive male rugby union players were recruited. Players were tested during three conditions: CONTROL (resistance training session only), CYCLE and RUN (corresponding to a concurrent training scheme with cycling or running sprint interval exercise conducted on the morning, followed by a resistance training session). Four hours rest was proposed between the aerobic and resistance training session. Muscle performance (bar velocity during bench press and box squat, counter movement jump height) and subjective ratings (rate of perceived exertion, wellbeing) were assessed during and after aerobic or resistance training sessions. No significant difference was observed for muscle performance (vertical jump height and bar velocity). However, significant higher perceived exertion and low-value scaled subjective wellbeing were observed in RUN (7.7 ± 1.1 and 17.9 ± 4.1, respectively) as compared with the two other conditions (6.7 ± 1.5 and 21.1 ± 3.6 for CONTROL and 7.4 ± 1.1 and 20.1 ± 3.9 for CYCLE). It was concluded that the exercise modality (running or cycling) during the aerobic exercise using a sprint interval exercise did not impact the quality of the resistance session. However, subjective ratings were affected the following days. Cycling exercises might be more adequate when performing a sprint interval training session during concurrent training programs.

## 1. Introduction

Concurrent training is the practice of endurance and strength modalities close to each other [1]. While these qualities are key factors for numerous sports, many studies have highlighted that the simultaneous training of both qualities could led to the blunting of some training adaptations. For instance, strength training increases the contractile capacity and muscle fiber size but decreases mitochondrial and capillary density and oxidative enzyme activity [2]. In contrast, aerobic training increases mitochondrial and capillary density and oxidative enzymes activity [3,4]. The origins of these interferences are multifactorial [1,5]. Two main hypotheses have been proposed in the literature [3]. Firstly, the acute hypothesis mostly refers to fatigue [6,7,8]. Briefly, fatigue induced by a previous aerobic session might reduce the total work of a strength exercise [9,10], which ultimately might impair neuromuscular adaptations [11]. Secondly, the chronic hypothesis proposes that strength and aerobic exercises share conflicting training adaptative mechanisms which might potentially reduce the overall physiological adaptations [3].

To mitigate these negative effects, several strategies have been implemented within concurrent training programming [12,13]. Amongst the potential strategies that might reduce fatigue and therefore interferences, training order, recovery duration or the type of aerobic exercise have frequently been considered [14,15]. The modality of the aerobic exercise (e.g., cycling vs. running) is an alternative interesting candidate [16,17]. Indeed, while bouts of cycling or running exercises have been shown to impair the subsequent strength training session [18,19,20,21], a lower amount of fatigue has been suggested in favor of the cycling modality [16,22]. These results could partly be explained by the contraction types. Indeed, plyometric contractions, during running, might cause greater muscle damage as compared with the concentric-dominant nature of cycling [23,24,25]. However, the comparison between cycling and running aerobic exercises during concurrent training are scarce and mainly apply to long-duration, low-intensity exercises [21]. Moreover, the effects of high-intensity interval exercises during concurrent training have been investigated in isolated conditions (i.e., either while running or while cycling) [16,20,22]. Indeed, to the best of our knowledge, the acute effects of running or cycling modalities while performing high-intensity interval exercises, such as sprint interval exercises, on the ability to maximally conduct a subsequent resistance training session are yet to be compared.

The aim of this research was to examine the acute effects of cycling or running sprint interval exercise on subsequent upper and lower body strength production capacity. Concurrent training was considered in the context of training twice a day (regularly used during the pre-season period). It was hypothesized that the plyometric nature of sprint interval running exercise would impair strength at a greater extent than cycling sprint interval exercise. Strength was considered as the ability to maximally perform a comprehensive resistance training session. Moreover, because perception of effort is nowadays considered as a limiting factor for performance and subsequent training rather than a simple marker of exercise intensity [26], subjective ratings were evaluated during the following days (24 h and 48 h post-loading).

## 2. Materials and Methods

### 2.1. Participants

Twenty-five competitive male rugby union players were recruited (age: 26 ± 3 years; height: 179.5 ± 4.6 cm; body mass: 83.4 ± 10.7 kg) (Figure 1). The sample size was calculated a priori using G * Power software (Ver 3.1.9.6, Dusseldorf, Germany). The following values were implemented to perform repeated measurements analysis of variances (ANOVA): *p* = 0.05, (1 − β) = 0.95 with an effect size of 0.59. The minimum number of participants required was 20. Participants had at least four years of rugby union training experience. Excluding the rugby union match every week, volunteers trained 6.8 ± 1.3 h per week (3.3 ± 0.4 h rugby and 3.5 ± 0.9 h strength training). The experiment was conducted during the off-season with volunteers maintaining their training habits. They were requested to reproduce the same training sessions during the experimental weeks and to avoid any physical exercise 48 h before each experimental session. Additionally, dietary supplements were prohibited during this period. Before participation, volunteers were fully informed of the purpose of the study and experimental procedure. All gave their informed consent for inclusion before their participation. The study was conducted in accordance with the Helsinki Declaration without any deviation from the protocol approved by the ethics committee for sport sciences research (CERSTAPS, IRB00012476-2020-30-11-74).

### 2.2. Experimental Design

Participants initially conducted a familiarization session one week before the experimental procedure started. It aimed to collect anthropometric data as well as their one repetition maximal (1 RM) during different exercises according to previous procedures [27]. Exercises were box squat, bench press, hip thruster, split squat, weighted pull-ups and over-head press. The 1 RM values permitted to calculate the loads to use during the experimental resistance exercises. Moreover, volunteers were familiarized with the training modalities applied during the subsequent experimental sessions.

After familiarization, volunteers performed three experimental sessions randomly presented and interspersed by seven days. Each experimental session lasted an entire day (Figure 2). The three experimental conditions were: (1) concurrent training composed of an aerobic session performed on an ergocycle and followed by a resistance exercise session (CYCLE); (2) concurrent training composed of a running aerobic session followed by a resistance exercise session (RUN); (3) a control condition only composed of a single resistance exercise session (CONTROL). Aerobic sessions during CYCLE and RUN were always performed in the morning. The resistance exercise session started 4 h after the end of the aerobic training session as per previous study [14]. Measurements were conducted during the aerobic and resistance sessions as well as the following two days (see below). During the experimental days, meals (breakfast, lunch and dinner) were standardized to avoid any confounding effect.

### 2.3. Training Procedures

Sprint interval exercises: Participants started with a standardized warm-up. It consisted in 5 min of hip mobility exercises followed by dynamic stretches on the lower limb, lunges, hip thrusters, calf raises and athletic drills. Participants then performed a sprint interval exercise that consisted in 3 sets of 6 maximal efforts of 6 s interspersed with 24 s of passive recovery, as per the previous study [28]. Two minutes rest were conducted between each set. According to the randomization, volunteers completed this session either on an ergocycle (Wattbike Pro, Nottingham, UK) or while running on an artificial turf for CYCLE or RUN, respectively. During CYCLE, the ergocycle was calibrated using the air flow brake at level 8 and the magnetically resistance on level 2. The peak power during each sprint was measured. During RUN, participants wore a Global Positioning System (GPS) unit capturing data at 16 Hz (Catapult OptimEye S7, Melbourne, Australia). GPS units were located between the scapulae in a customized pocket placed in volunteers’ shirt. They were used to measure the peak speed achieved in each sprint. To limit potential inter-unit variability, each player wore the same unit for the total duration of the study. Fatigue within the aerobic session was evaluated as the percentage changes of the averaged peak power (CYCLE) and peak speed (RUN) between the first and last set.

Resistance exercise session: Participants started with a standardized warm-up. It included dynamic full body stretches and 3 sets of 3 repetitions on box squat and bench press at 50%, 70% and 80% of their 1 RM. Then, participants performed 4 counter movement jumps (CMJs) with a broomstick placed behind the neck to limit potential arm swings. The average vertical jump height was measured using a GymAware linear encoder (GymAware Power Tool, kinetic Performance Technologies, Canberra, Australia). Subsequently, participants completed the resistance exercise session. Players performed 4 sets of 6 maximum repetitions (6 RM) on the box squat then on the bench press. Each set included the main exercises (squat box or bench press) directly followed by 2 complementary exercises (hip thruster and split squat or weighted pull-ups and over-head press exercises). The lower body was always trained first and was followed by the upper body. Two minutes passive rest was allowed between sets. Three minutes of rest was allowed between the lower and upper body. For each repetition, participants were instructed to move the load as fast as possible during the concentric phase of the movement. The concentric peak velocity (PV; m.s^−1^) and concentric mean velocity (MV; m.s^−1^) were measured during each repetition of the box squat and bench press using the GymAware system. The averaged PV and MV during the whole resistance exercise session were calculated. Moreover, fatigue within the resistance exercise session was evaluated as the relative changes of the averaged PV and MV between the first and last set.

### 2.4. Subjective Outcomes

The rate of perceived exertion (RPE) was evaluated using a 10-point scale (10 being the greatest perceived exertion). Data were collected into GoogleForm© 30 min after each aerobic and resistance training session. Participants also had to rate their subjective wellbeing the next two days (D + 1, D + 2). A 6-item questionnaire adapted from Mclean et al. [29] was used to rate sleep, nervousness, fatigue, muscle soreness, palpation soreness and downstairs soreness on a 5-point Likert scale. Each item was rated from 1 to 5 (5 corresponding to a good sleep, low nervous nervousness, low fatigue and low soreness). The overall wellbeing was assessed by summing all 6 scores. A high score corresponded to a good state of wellbeing. Subjects completed the questionnaire on their own in order to prevent any influence from other players [30].

### 2.5. Statistical Analyses

Data are presented as mean values ± standard deviation. The aerobic session was tested using a student *t*-test for RPE and percentage changes of peak power and peak velocity between the first and last set during CYCLE and RUN, respectively. Vertical jump height, averaged MV and PV during the whole session, relative MV and PV changes throughout the session and RPE after the resistance exercise session were analyzed using a one-way analysis of variances (ANOVA). The condition (CYCLE vs. RUN vs. CONTROL) was used as repeated measures. The averaged MV and PV during the first and last series were also tested using a two-way ANOVA (condition × time). Time corresponded to the comparison between the first and last series. A two-way (condition × time) ANOVA was also conducted for the 6-item questionnaire. In that case, time corresponded to D + 1 and D + 2. In case of significant main effects or interactions, post hoc analyses with Bonferroni corrections were conducted. The statistical significance was accepted for *p* < 0.05. The assumptions for repeated-measures ANOVA (sphericity and normal distribution) were tested using Mauchly and Shapiro–Wilk tests. The effect sizes were calculated for all analyses. Partial-eta-squared (pη^2^) were calculated from ANOVA results, with values of 0.01, 0.06 and above 0.14 representing small, medium and large differences, respectively. Subsequently, qualitative descriptors of standardized effects were used for pairwise comparisons with Cohen’s d < 0.5, 0.5–1.2 and >1.2, representing small, medium and large magnitudes of change, respectively [31]. Statistics were conducted using JASP (Ver 0.13, JASP Team (2020), University of Amsterdam, Amsterdam, The Netherlands).

## 3. Results

A 15.8 ± 8.6% and 6.6 ± 5.2% decrease in peak power and peak velocity was obtained during cycling and running aerobic exercises, respectively. This fatigue was significantly more pronounced during cycling as compared with running (Table 1). At the end of the two aerobic sessions, similar RPE values were obtained (7.5 ± 1.0 and 7.7 ± 1.2 after cycling and running, respectively).

Vertical jump heights measured before the resistance exercise sessions were not different between conditions (Table 1). The averaged values of MV and PV during the whole resistance exercise session were not significantly different between conditions for both the box squat and bench press (Table 1). When considering MV and PV during the first and last sets, the ANOVA revealed a significant main time effect during both the box squat and bench press exercises (Table 1). No main condition effect or interaction (condition x time) was obtained (see Table 1 and Figure 3). For box squat, post hoc analyses revealed that PV and MV significantly decreased during the resistance exercise session (−5.0 ± 9.9%, *p* = 0.012, d = 0.545 and −5.5 ± 11.4%, *p* = 0.012, d = 0.542, respectively) (Figure 4). For bench press, post hoc analyses revealed that PV and MV significantly decreased during the resistance exercise session (−9.3 ± 13.3%, *p* = 0.001, d = 0.748 and −13.5 ± 15.9%, *p* < 0.001, d = 0.900, respectively) (Figure 4). These percentage changes were not different between conditions (Table 1).

RPE values measured at the end of the resistance exercise session were significantly different between conditions (Table 1). Post hoc analyses revealed that RPE was significantly lower (*p* = 0.042, d = 0.510) following the CONTROL condition (6.7 ± 1.5) as compared with RUN (7.7 ± 1.1). RPE following the resistance exercise session in the CYCLE condition (7.4 ± 1.1) was not different than CONTROL or RUN (*p* = 0.131, d = 0.196 and *p* = 1.000, d = 0.133, respectively). Of the volunteers, 20 and 18 out of 25 had lower RPE in the CONTROL condition as compared with RUN and CYCLE, respectively. The 6-item questionnaire presented on D + 1 and D + 2 revealed a main condition effect for the overall score, fatigue and overall soreness (Table 2). A main time factor effect was obtained for fatigue and overall soreness. No condition × time interaction was observed. Briefly, the overall score was significantly lower in the RUN condition as compared with CYCLE and CONTROL (*p* = 0.002, d = 0.744 and *p* < 0.001, d = 1.037, respectively), without any difference between CYCLE and CONTROL (*p* = 0.446, d = 0.294). Similar effects were observed for muscle soreness, soreness at palpation and downstairs soreness. In contrast, fatigue was significantly greater (low score) in RUN as compared with CONTROL (*p* = 0.015, d = 0.589), without any difference between CYCLE and RUN, and between CYCLE and CONTROL (*p* = 0.400, d = 0.305 and *p* = 0.488, d = 0.283, respectively) (Table 3). Post hoc analyses conducted on the time effect revealed that fatigue and muscle, palpation and downstairs soreness recovered between D + 1 and D + 2 (*p* < 0.001, d = 0.886; *p* < 0.001, d = 0.985; *p* = 0.015, d = 0.528 and *p* = 0.002, d = 0.677, respectively).

## 4. Discussion

The aim of the present study was to assess the acute effects of sprint interval cycling or running on subsequent upper and lower body strength capacities in the context of concurrent training. It was hypothesized that running sprint interval exercise would impair strength at a greater extent than cycling sprint interval exercise. The findings of this study partially confirmed this hypothesis. Indeed, the modality of the aerobic exercise had no impact on strength capacities during the resistance exercise session. However, concurrent training with a running aerobic exercise produced greater subjective fatigue and soreness as compared with a concurrent training associated with a cycling aerobic exercise.

The main result of the present study was that the overall strength capacities were similar between our three experimental sessions. It suggested that during concurrent training the use of sprint interval training (either while running or cycling) has no impact on the subsequent resistance exercise session. This conclusion was firstly witnessed by the vertical jump heights. During our concurrent training, four hours rest were permitted between the aerobic and resistance exercise sessions. Because vertical jumps were similar at the onset of all resistance exercise sessions (also similar than the control condition), it seems that the delay was enough to alleviate the likely fatigue resulting from the aerobic exercises. Secondly, our conclusion was endorsed by the similar mean and peak velocities registered during the whole resistance exercise session as well as similar fatigue measured throughout the session for the three experimental conditions. Slightly longer recovery delays were previously recommended between two training sessions in order to minimize the negative effect of neuromuscular fatigue on the subsequent resistance exercise sessions [14,32]. However, in a recent study, authors concluded that fatigue was not exacerbated when a resistance exercise session was performed only one hour after an intermittent sprint running exercise [33]. Accordingly, during concurrent training using high-intensity interval training, short rest durations (at least four hours as programmed here) could be proposed between aerobic and resistance exercise sessions without any detrimental fatigue. Moreover, performing cycling or running aerobic exercise with the present sprint interval exercise design has no mechanical impact on a subsequent resistance training session when individuals train twice a day.

The minimal delay between aerobic and resistance exercise sessions should be further explored. Beside some authors suggested one hour was enough between sessions [33], this conclusion could not be generalized. Indeed, although the time course of fatigue was not explored here, it seems reasonable to argue that cycling or running exercises would have produce different acute fatigue. Our results confirmed this hypothesis since a greater fatigue was obtained during cycling aerobic training session as compared with running. Such finding is in general accordance with the literature [28,34]. For instance, some authors [28] investigated the effects of repeated sprint ability (5 × 6 s sprints) while cycling or running. A larger decrease in sprint performance was obtained across the procedure after cycling than running. The difference between the two modes of locomotion during these sprint exercises was attributed to the larger fractional duration of the external force during cycling. Indeed, the duty cycle (fraction of the total sprint time that involves external force application by a single limb to the pedal or ground) is twice greater (0.50 vs. 0.24) while cycling than running and therefore impact fatigue time course [35].

In contrast to the different mechanical fatigue registered during the two modes of locomotion, volunteers reported similar RPE values after cycling or running sprint exercises. This result disagrees with previous studies [28,35]. Greater RPE values were generally obtained following repeated cycling sprints. The conflicting result we obtained could be attributed to the procedure used (three times more sprints in our study) or to the training status of the volunteers under investigation. Indeed, regularly trained and competitive volunteers were considered in our study, while moderately trained individuals were evaluated by Rampinini et al. [28]. It can be speculated that, for well-trained individuals, perception of effort was not dependent on the mode of locomotion but was mostly affected by the type of effort.

As previously concluded, sprint interval training sessions, whether performed using a cycling or running locomotion mode, has no mechanical impact on a subsequent resistance training session. This conclusion is in contradiction with our a priori hypothesis but in accordance with recent published papers [36,37]. Indeed, cycling and running differ with respect to muscle contraction types. Considering stretch-shortening cycle during running exercises (with a prevalent eccentric component as compared with the purely concentric cycling actions), larger influences during the resistance exercise session were expected. Indeed, it is well known that the physiological demands, mechanical and recruitment pattern of the motor units diverge between concentric and eccentric conditions [38,39,40]. Moreover, a running aerobic session that involve significant eccentric loading would induce greater muscle strain, fatigue and soreness than following concentric contractions [41,42,43]. The so-obtained fatigue and potential concomitant muscle damage would influence force output several hours after the exercise [44,45]. Interestingly, beside a lack of difference for mechanical output between our three experimental conditions, perception revealed some effects that could have large impact for athletes. RPE after the resistance exercise session and the overall wellbeing score or soreness the following days were negatively impacted in the running condition (as compared with both cycling and control conditions). Running might therefore negatively influence physical or technical training that could be performed during the subsequent days. This is potentially due to the different kinetics of subjective ratings and especially perceived soreness compared with performance outcomes and/or inflammatory markers [46]. These lower subjective ratings following running concurrent training are in accordance with our initial hypothesis. They are consistent with the likely larger damage induced by running than cycling. Running aerobic sprint interval training session would therefore be less efficient during concurrent training. Such conclusion must be considered by coaches when programming their training week by taking into account the likely physiological and perceptual lag induced by a running session.

## 5. Limitations

Some limitations such as the lack of information related to the etiology of fatigue (i.e., peripheral vs. central) and muscle damage should be acknowledged. Measuring the time course of the neuromuscular fatigue would have shed some light on the complexity of concurrent training responses. Moreover, heart rate monitoring during endurance sessions could have brought more insights into players internal load during both running and cycling conditions. Finally, only one format of exercise (repeated sprint) was used in the present study. Considering the likely differences in energy supply, it is possible that another form of sprint interval exercise would lead to different conclusions. Future studies should assess the acute effects of different sprint interval exercise format and their respective neuromuscular fatigue responses on a subsequent resistance training session.

## 6. Conclusions

The present study revealed that during concurrent training, the exercise modality (running vs. cycling) during the sprint interval exercise (when performed first) has no effect on mechanical output during the subsequent resistance exercise session but has an influence on subjective ratings. Performing cycling aerobic exercises seemed preferable when training should be repeated (for example during pre-season periodization). Moreover, our results highlighted that training sessions could be performed with a short delay between aerobic and resistance exercise sessions. Future studies should be conducted to evaluate the chronic effects of these different modalities on the neuromuscular system.

## Figures and Tables

**Figure 1 sports-10-00075-f001:**
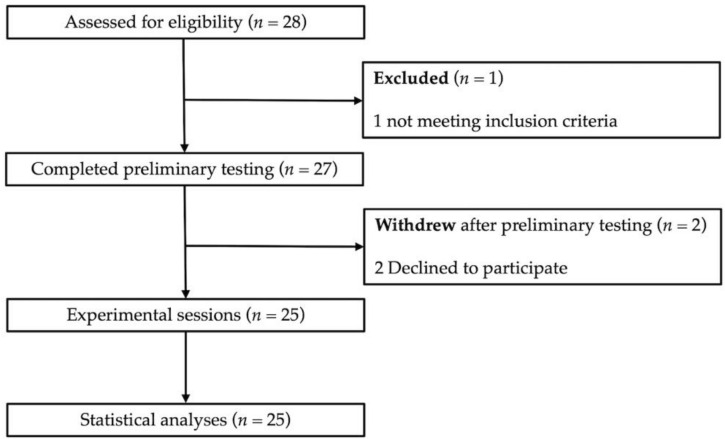
Study flowchart.

**Figure 2 sports-10-00075-f002:**
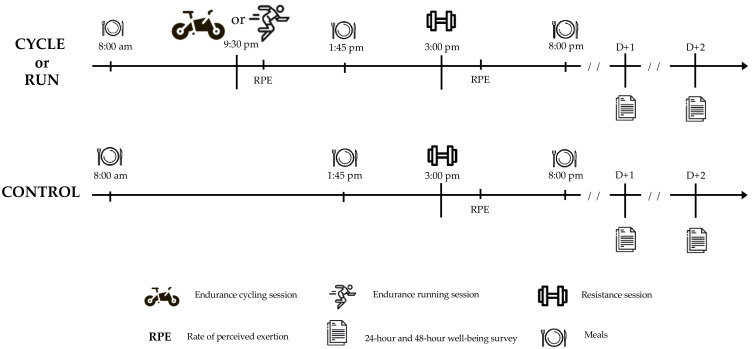
Schematic representation of the three experimental sessions. CYCLE and RUN are concurrent training conditions composed of a cycling or running aerobic session using a sprint interval exercise followed by a resistance exercise training session. The CONTROL condition only included the same resistance training session.

**Figure 3 sports-10-00075-f003:**
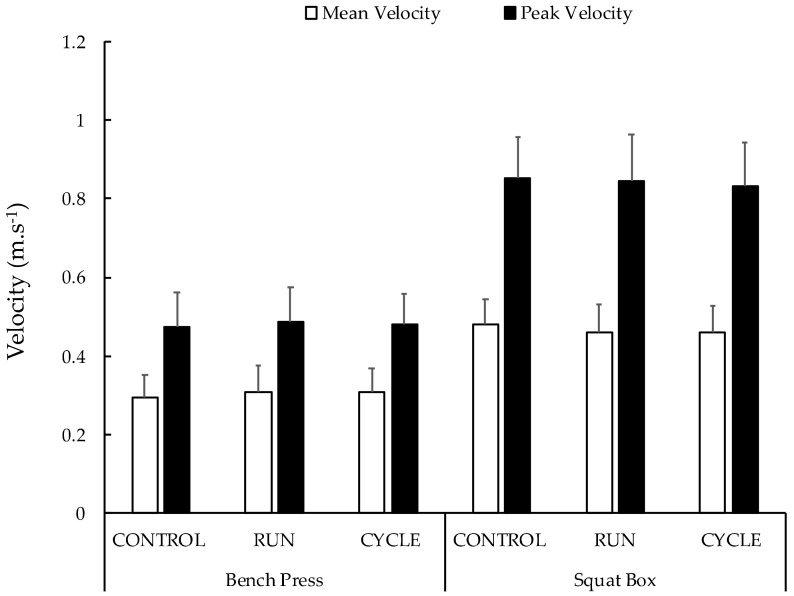
Mean velocity and peak velocity during the three experimental conditions on the bench press and squat box exercises. Values represented the averaged velocities during the whole resistance exercise session ± standard deviation. No significant differences were observed between conditions.

**Figure 4 sports-10-00075-f004:**
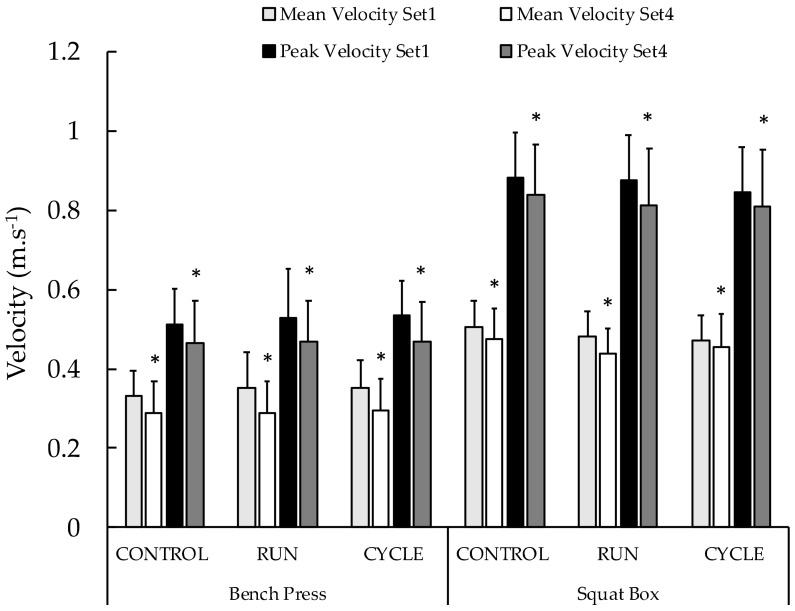
Mean velocity and peak velocity between the first (Set 1) and the last set (Set 4) of the three experimental conditions for the bench press and squat box exercises. *—significant differences between the first and last set for the two exercises and for both the mean and peak velocities (*p* < 0.05).

**Table 1 sports-10-00075-t001:** Results of the *t*-tests and ANOVA for all outcomes during the endurance and strength sessions.

Session/Exercise	Variable	Condition Factor		Time Factor		Condition × Time	
		*p*-Value	d/pη^2^	*p*-Value	pη^2^	*p*-Value	pη^2^
Endurance session	RPE at the end of the session	0.519	−0.131	-	-	-	-
	% change peak power vs. peak velocity	<0.001 *	0.789	-	-	-	-
Box squat velocity	MV during the whole session	0.308	0.048	-	-	-	-
	MV during the first and last series	0.088	0.096	0.012 *	0.234	0.336	0.044
	% change of the MV	0.334	0.045	-	-	-	-
	PV during the whole session	0.494	0.029	-	-	-	-
	PV during the first and last series	0.274	0.052	0.012 *	0.236	0.626	0.019
	% change of the PV	0.615	0.020	-	-	-	-
Bench press velocity	MV during the whole session	0.436	0.034	-	-	-	-
	MV during the first and last series	0.716	0.014	0.001 *	0.368	0.758	0.012
	% change of the MV	0.877	0.05	-	-	-	-
	PV during the whole session	0.658	0.017	-	-	-	-
	PV during the first and last series	0.598	0.021	<0.001 *	0.457	0.720	0.014
	% change of the PV	0.971	0.001	-	-	-	-
Resistance exercise session	Vertical jump height before the session	0.089	0.096	-	-	-	-
	RPE at the end of the session	0.038 *	0.127	-	-	-	-

MV—mean velocity; PV—peak velocity; RPE—rate of perceived exertion. *p*-values and effect sizes (d or pη^2^) are shown. *—significant main effects or interactions (*p* < 0.05).

**Table 2 sports-10-00075-t002:** Results of the ANOVA for the wellbeing questionnaire.

Variable	Condition Factor		Time Factor		Condition × Time	
	*p*-Value	pη^2^	*p*-Value	pη^2^	*p*-Value	pη^2^
Overall score	<0.001 *	0.373	0.100	0.109	0.984	0.000
Sleep	0.369	0.041	0.913	0.000	0.957	0.002
Nervousness	0.445	0.033	0.252	0.054	0.978	0.000
Fatigue	0.019 *	0.153	<0.001 *	0.450	0.741	0.012
Muscle soreness	0.006 *	0.190	0.001 *	0.503	0.892	0.005
Palpation soreness	0.008 *	0.180	0.015 *	0.222	0.704	0.015
Downstairs soreness	0.003 *	0.218	0.002 *	0.323	0.213	0.062

*p*-values and effect sizes (pη^2^) are shown. *—significant main effects or interactions (*p* < 0.05).

**Table 3 sports-10-00075-t003:** Results from the 6-item wellbeing questionnaire.

Days	Conditions	Sleep	Nervous	Fatigue €	Muscle Soreness €	Palpations Soreness €	DownStairs Soreness €	Overall Score
D + 1	CYCLE	3.9 ± 0.8	3.5 ± 0.8	2.7 ± 1.0	2.7 ± 1.0	3.5 ± 0.8	3.7 ± 0.8	20.1 ± 3.9
	RUN	3.9 ± 1.0	3.6 ± 0.9	2.5 ± 0.9 $	2.1 ± 1.1 *	2.9 ± 0.8 *	3.0 ± 0.9 *	17.9 ± 4.1 *
	CONTROL	4.1 ± 0.8	3.7 ± 0.8	3.1 ± 0.9	2.9 ± 1.1	3.5 ± 0.8	3.7± 0.8	21.1 ± 3.6
D + 2	CYCLE	4.0 ± 0.6	3.6 ± 1.0	3.2 ± 1.0	3.2 ± 1.1	3.7 ± 0.9	3.9 ± 0.8	21.7 ± 4.0
	RUN	3.9 ± 1.1	3.4 ± 0.9	2.9 ± 1.0 $	2.5 ± 1.1 *	3.3 ± 0.8 *	3.5 ± 1.3 *	19.4 ± 4.0 *
	CONTROL	4.1 ± 0.8	3.7 ± 0.8	3.4 ± 1.0	3.3 ± 1.2	3.7 ± 1.0	4.2 ± 0.8	22.5 ± 4.6

Mean values ± standard deviation. D + 1 and D + 2 are results obtained 24 h and 48 h after the end of the experimental sessions. *—Significantly different with CYCLE and CONTROL (*p* < 0.05). $—Significantly different with CONTROL (*p* < 0.05). €—Significant differences between D + 1 and D + 2 (*p* < 0.05).

## Data Availability

The authors declare that the dataset is available on request.

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
