# Peer review of "Concurrent Training Programming: The Acute Effects of Sprint Interval Exercise on the Subsequent Strength Training"

_sports, 2022, doi:10.3390/sports10050075_

Round 1

Reviewer 1 Report

Dear authors

I must compliment the effort speeded in this research. Although the concurrent training is still under the attention of many researchers ( hundreds of papers in the last 10 years), there is some novelty in the current approach that the reviewer claims the authors to highlight in the introduction section.

The reviewer points to other aspects that could improve the paper.

The abstract needs to include some data, not only almost conclusions.

When reporting statistical power, additional information related to the statistical analysis the authors want to use should be added (briefly, the family and statistical test).

The authors mention that the three experiments were interspersed over seven days to ensure the same physiological status in the experimental design. It means a training loss for three weeks ( not considering familiarisation time). Could the authors please detail the training schedule during this long period?

We assume that the two minutes of rest between sets were also passive in the training procedures. Correct?

In the resistance training session, the authors assume that the lower body was always trained first and was followed by the upper body. Could the authors, please, add the rationale to this option?

Results are well presented, including graphs and tables.

The discussion section is one important part of the paper, addressing the main topics covered.

The reviewer suggests the authors amplify the discussion section, including some recently published papers

ex.Petré H, Hemmingsson E, Rosdahl H, Psilander N. Development of Maximal Dynamic Strength During Concurrent Resistance and Endurance Training in Untrained, Moderately Trained, and Trained Individuals: A Systematic Review and Meta-analysis. Sports Med. 2021 May;51(5):991-1010. doi: 10.1007/s40279-021-01426-9. Epub 2021 Mar 22. PMID: 33751469; PMCID: PMC8053170.

Murlasits Z, Kneffel Z, Thalib L. The physiological effects of concurrent strength and endurance training sequence: A systematic review and meta-analysis. J Sports Sci. 2018 Jun;36(11):1212-1219. doi: 10.1080/02640414.2017.1364405. Epub 2017 Aug 7. PMID: 28783467.

and others.

Author Response

Dear authors

I must compliment the effort speeded in this research. Although the concurrent training is still under the attention of many researchers (hundreds of papers in the last 10 years), there is some novelty in the current approach that the reviewer claims the authors to highlight in the introduction section.

Response:

Dear reviewer.

Thank you for taking the time to review this paper. We also thank the reviewer for the substantial work done and the suggestions that will surely help improve the manuscript. We did our best to take all specific comments into account. All the comments have been addressed and appear as track changes in the text.

The reviewer points to other aspects that could improve the paper.

The abstract needs to include some data, not only almost conclusions.

Response: Thank you for your suggestion. We added more information related to the results. Please see the changes in abstract: "No significant difference was observed for muscle performance (vertical jump height and bar velocity). However, significant higher perceived exertion and low values scale subjective wellbeing were observed in RUN (7.7 ± 1.1 and 17.9 ± 4.1, respectively) as compared to the two other conditions (6.7 ± 1.5 and 21.1 ± 3.6 for CONTROL and 7.4 ± 1.1 and 20.1 ± 3.9 for CYCLE)."

When reporting statistical power, additional information related to the statistical analysis the authors want to use should be added (briefly, the family and statistical test).

Response: We added the statistical model (i.e. repeated measurements ANOVA) used during the a priori sample size determination. Please see the following the chapter 2.1 participants: "The following values were implemented to perform repeated measurements analysis of variances (ANOVA): p = 0.05, (1-β) = 0.95 with an effect size of 0.59."

The authors mention that the three experiments were interspersed over seven days to ensure the same physiological status in the experimental design. It means a training loss for three weeks (not considering familiarisation time). Could the authors please detail the training schedule during this long period?

Response: The experiment was conducted during the off-season. Individuals could train but they were asked to reproduce the same trainings during the experimental weeks. This part of the experimental design has been detailed for clarity. Please see the following the chapter 2.1 participants: "The experiment was conducted during the off-season with volunteers maintaining their training habits. They were requested to reproduce the same training sessions during the experimental weeks and to avoid any physical exercise 48 hours before each experimental session."

We assume that the two minutes of rest between sets were also passive in the training procedures. Correct?

Response: This is correct. Thank you for pointing this out. We added this information in chapter 2.3 training session procedures: about the recovery modality line 171: "Two minutes passive rest was allowed between sets"

In the resistance training session, the authors assume that the lower body was always trained first and was followed by the upper body. Could the authors, please, add the rationale to this option?

Response: Thank you very much for this question. The rationale for training the lower-body first was related to the recommendations of the American of Sport college of medicine: Ratamess et al.  Progression Models in Resistance Training for Healthy Adults. Med. Sci. Sport. Exerc. 2009, 41 (3), 687–708. https://doi.org/10.1249/MSS.0b013e3181915670.

Results are well presented, including graphs and tables.

Response: Thank you for this comment.

The discussion section is one important part of the paper, addressing the main topics covered.

The reviewer suggests the authors amplify the discussion section, including some recently published papers

ex.Petré H, Hemmingsson E, Rosdahl H, Psilander N. Development of Maximal Dynamic Strength During Concurrent Resistance and Endurance Training in Untrained, Moderately Trained, and Trained Individuals: A Systematic Review and Meta-analysis. Sports Med. 2021 May;51(5):991-1010. doi: 10.1007/s40279-021-01426-9. Epub 2021 Mar 22. PMID: 33751469; PMCID: PMC8053170.

Murlasits Z, Kneffel Z, Thalib L. The physiological effects of concurrent strength and endurance training sequence: A systematic review and meta-analysis. J Sports Sci. 2018 Jun;36(11):1212-1219. doi: 10.1080/02640414.2017.1364405. Epub 2017 Aug 7. PMID: 28783467.

and others.

Response: Thank you very much for this suggestion. We added these references in the last paragraph of the discussion just before the limitations section

Reviewer 2 Report

The term training would suggest a structured exercise program over a period of time. This would normally be characterised by changes in overload and progression. However, the study provides observations on the acute exercise responses. I suggest to reconsider the terminology throughout the paper to reflect that.

L21. Please clarify what the differences were for RUN condition. Were the changes beneficial?

Please strengthen the introduction by providing a rationale for examining the subjective and overall well-being parameters.

Please clarify whether the experimental conditions were done in what part of the competitive season (e.g. pre-, mid- or post-season) as it looks the participants were not allowed any other exercise sessions for a number of weeks.

The cycling and running sessions are described as endurance exercise sessions and aerobic sessions, this would suggested moderate-intensity continuous exercise. However, the cycling and running were executed with maximal effort (all out pacing) so should they not be better described as sprint interval exercises. High intensity interval training is terminology more applicable to exercise bouts of minutes (not all out pacing) with rest periods of minutes. Please reconsider/revise throughout the manuscript.

L29. High-intensity resistance training may develop aerobic and strength qualities but is not considered concurrent training. I suggest to clarify that concurrent training is the practice of two different training modalities close to each other, e.g. moderate-intensity continuous exercise and resistance exercise as part of a training program (as was done in the Hickson study) with indeed the aim to develop different adaptations. I suggest to rephrase what is considered concurrent training.

L31. I suggest to change “physiological interferences” to “the blunting of some training adaptations”. It is the case that subsequent statements are about adaptations.

Ls 46-51. There is mention of previous studies examining effects of aerobic exercise on strength exercise responses. I suggest to mention briefly that the intensity and duration of the aerobic exercise modalities in those studies will be different that used in the present study. That should bring out more the novelty of the study.

L74. Physical activity is considered any bodily movement that requires energy expenditure. Do you mean that structured physical exercise was to be avoided. Please clarify.

L89. I suggest to change “experimental resistance” to “experimental resistance exercise”.

L94. How can the authors be sure that seven days between sessions ensure similar physiological status”. Was that quantified with physiological parameters? Please clarify.

L96 and throughout the paper. I suggest to replace “resistance session” with “resistance exercise session”.

L113. Ref 22 has not the exercise as described here. Please revise.

L118. I suggest to present data on peak power during the cycling and peak speed during the running.

L160. Change “power or velocity” to “peak power and peak velocity”.

Ls 214. RPE is recorded as a whole number (without decimals) but then the mean and SD of the group RPEs expressed with one decimal place. Is that allowed? I suggest to provide individual responses, i.e. how many participants had a lower RPE value in the CONTROL condition.

Ls 269. “Moreover, performing cycling or running aerobic exercises has no mechanical impact on a subsequent resistance training session when individuals train twice a day.”. Please clarify that it was specific to the parameters of the cycling and running exercises in the present study.

L286. Is blood flow an issue during dynamic activities as Ref 35 is on prolonged isometric contractions. Please reconsider/revise. In addition, Ref 36 is on blood flow restriction, and again this is chronic restriction. Please reconsider/revise.

L312. How meaningful is what it seems a difference of 1 on the Borg scale?

L339. Please change “locomotion” and that is not a description for stationary cycling.

Ref 26 and 33 have no doi.

Author Response

Response: We thank both reviewers for taking the time to review this paper. We also thank the reviewer for the substantial work done and the suggestions that will surely help improve the manuscript. We did our best to take all specific comments into account. All the comments have been addressed and appear as track changes in the text.

The term training would suggest a structured exercise program over a period of time. This would normally be characterised by changes in overload and progression. However, the study provides observations on the acute exercise responses. I suggest to reconsider the terminology throughout the paper to reflect that

Response: Thank you very much for your comment. We do understand your point and we agree with you. Throughout this manuscript we have tried as much as possible to make it clearer. We regularly add the word session to clarify it was related to a training session as opposed to (chronic) training program.

L21. Please clarify what the differences were for RUN condition. Were the changes beneficial? Response: Thank you very much for your comment. We amended this part according to your comment. Please see the change in abstract: "No significant difference was observed for muscle performance (vertical jump height and bar velocity). However, significant higher perceived exertion and low values scale subjective wellbeing were observed in RUN (7.7 ± 1.1 and 17.9 ± 4.1, respectively) as compared to the two other conditions (6.7 ± 1.5 and 21.1 ± 3.6 for CONTROL and 7.4 ± 1.1 and 20.1 ± 3.9 for CYCLE)."

Please strengthen the introduction by providing a rationale for examining the subjective and overall well-being parameters.

Response: perceived exertion and subjective ratings are considered as limiting factors for subsequent performance and not just a marker of fatigue or exercise intensity. Accordingly, these outcomes are very important. We therefore added one reference in introduction and slightly modified the last sentence in introduction to justify this important measurement: "Moreover, because perception of effort is nowadays considered as a limiting factor for performance and subsequent training rather than a simple marker of exercise intensity [27], subjective ratings were evaluated during the following days (24 and 48 h post-loading)."

Please clarify whether the experimental conditions were done in what part of the competitive season (e.g. pre-, mid- or post-season) as it looks the participants were not allowed any other exercise sessions for a number of weeks.

Response: The experiment was conducted during the off-season. Individuals could train but they were asked to reproduce the same trainings during the experimental weeks. This part of the experimental design has been detailed for clarity. Please see the following the chapter 2.1 participants: "The experiment was conducted during the off-season with volunteers maintaining their training habits. They were requested to reproduce the same training sessions during the experimental weeks and to avoid any physical exercise 48 hours before each experimental session."

The cycling and running sessions are described as endurance exercise sessions and aerobic sessions, this would suggest moderate-intensity continuous exercise. However, the cycling and running were executed with maximal effort (all out pacing) so should they not be better described as sprint interval exercises. High intensity interval training is terminology more applicable to exercise bouts of minutes (not all out pacing) with rest periods of minutes. Please reconsider/revise throughout the manuscript.

Response: Thank you very much for this comment. We previously indicated in abstract or in introduction it was a high-intensity exercise. However, according to your comment, and for clarity, we regularly added the information it was a sprint interval exercise. Please see changes within the manuscript.

L29. High-intensity resistance training may develop aerobic and strength qualities but is not considered concurrent training. I suggest to clarify that concurrent training is the practice of two different training modalities close to each other, e.g. moderate-intensity continuous exercise and resistance exercise as part of a training program (as was done in the Hickson study) with indeed the aim to develop different adaptations. I suggest to rephrase what is considered concurrent training.

Response: We agree with reviewer's comment. This sentence has been modified accordingly. Please see the first sentence of the introduction: "Concurrent training is the practice of endurance and strength modalities close to each other [1]."

L31. I suggest changing “physiological interferences” to “the blunting of some training adaptations”. It is the case that subsequent statements are about adaptations.

Response: We agree with reviewer's comment. This sentence has been modified accordingly. Please see the second sentence of the introduction: "While these qualities are key factors for numerous sports, many studies have highlighted that the simultaneous training of both qualities could led to the blunting of some training adaptations."

Ls 46-51. There is mention of previous studies examining effects of aerobic exercise on strength exercise responses. I suggest to mention briefly that the intensity and duration of the aerobic exercise modalities in those studies will be different that used in the present study. That should bring out more the novelty of the study.

Response: According to your comment, the end of the second paragraph of the introduction has been modified to clarify the aerobic procedures previously investigated: "However, the comparison between cycling and running aerobic exercises during concurrent training are scarce and mainly apply long duration low intensity exercises [21]. Moreover, the effects of high-intensity interval exercises during concurrent training have been investigated in isolated conditions (i.e., either while running or while cycling) [20,22,23]. Indeed, to the best of our knowledge, the acute effects of running or cycling modalities while performing high-intensity interval exercises such as sprint-interval exercises on the ability to maximally conduct a subsequent resistance training session are yet to be compared."

L74. Physical activity is considered any bodily movement that requires energy expenditure. Do you mean that structured physical exercise was to be avoided. Please clarify.

Response: As indicated in a previous response, individuals could train but they were asked to reproduce the same trainings during the experimental weeks and to avoid any physical exercise 48 hours before all experimental sessions. This part of the experimental design has been detailed for clarity. Please see the following the chapter 2.1 participants: "The experiment was conducted during the off-season with volunteers maintaining their training habits. They were requested to reproduce the same training sessions during the experimental weeks and to avoid any physical exercise 48 hours before each experimental session."

L89. I suggest to change “experimental resistance” to “experimental resistance exercise”.

Response: The sentence has been modified according to your comment: "The 1RM values permitted to calculate the loads to use during the experimental resistance exercises."

L94. How can the authors be sure that seven days between sessions ensure similar physiological status”. Was that quantified with physiological parameters? Please clarify.

Response: Thank you for your comment. As indicated previously, the experiment was conducted during the off-season and participants were requested to avoid physical activity 48h before each experiment session to limit any bias. However, as suggested by your comment, no direct evaluation permitted to ensure similar physiological status for all three sessions. Accordingly, this sentence has been deleted.

L96 and throughout the paper. I suggest to replace “resistance session” with “resistance exercise session”.

Response: Thank you for this suggestion. We amended accordingly. Please see the changes throughout the paper.

L113. Ref 22 has not the exercise as described here. Please revise.

Response: As indicated by the reviewer, reference 22 was incorrectly cited. We corrected and cited ref 33 that used the same training pattern (6 s efforts with 24 s recovery). Please see the very beginning of chapter 2.3 training procedures: "Participants then performed a sprint interval exercise that consisted in 3 sets of 6 maximal efforts of 6 seconds interspersed with 24 seconds of passive recovery as per previous study [33]."

L118. I suggest to present data on peak power during the cycling and peak speed during the running.

Response: Initially we wanted to present these data. However, we decided not to present peak power and peak velocity data for 2 main reasons. First, we have numerous tables and figures. Secondly, we don't think these data are helpful for the study understanding. Indeed, power and velocity have different units and applying a normalisation procedure would be similar than the % changes (presented here). Accordingly, we have not taken into account your comment and we have not modified the manuscript. 

L160. Change “power or velocity” to “peak power and peak velocity”.

Response: This sentence has been modified according to your comment. Please see chapter 2.5 statistical analysis: "The aerobic session was tested using a student t-test for RPE and percentage changes of peak power and peak velocity between the first and last set during CYCLE and RUN, respectively."

Ls 214. RPE is recorded as a whole number (without decimals) but then the mean and SD of the group RPEs expressed with one decimal place. Is that allowed? I suggest to provide individual responses, i.e. how many participants had a lower RPE value in the CONTROL condition.

Response: RPE is often presented with decimals in number or in graphs (See Marcora S papers; for example: https://doi.org/10.3389/fnhum.2015.00067). 18 volunteers had higher RPE in RUN as compared to CONTROL. This information has been added in results section: "Twenty volunteers and 18 out of 25 had lower RPE in CONTROL condition as compared to RUN and CYCLE, respectively. "

Ls 269. “Moreover, performing cycling or running aerobic exercises has no mechanical impact on a subsequent resistance training session when individuals train twice a day.”. Please clarify that it was specific to the parameters of the cycling and running exercises in the present study.

Response: According to your comment, we now indicated in this sentence that the conclusion is specific to our sprint-interval exercise. Please see the discussion last sentence of the second paragraph: "Moreover, performing cycling or running aerobic with the present sprint-interval exercise design, has no mechanical impact on a subsequent resistance training session when individuals train twice a day."

L286. Is blood flow an issue during dynamic activities as Ref 35 is on prolonged isometric contractions. Please reconsider/revise. In addition, Ref 36 is on blood flow restriction, and again this is chronic restriction. Please reconsider/revise.

Response: As suggested by your comment, this sentence was unclear and also useless. Accordingly, this sentence was deleted.

L312. How meaningful is what it seems a difference of 1 on the Borg scale?

Response: A difference of 1 on the Borg scale could appear low but has large impact on performance. Considering the Borg scale 6.7 is referred as "hard" while 7.7 is "really hard". With RPE lower than 6, a 1 point difference would not have any impact but at these levels, it has.

L339. Please change “locomotion” and that is not a description for stationary cycling.

Response: This sentence has been modified according to your comment. Please see the first sentence of the conclusion: "The present study revealed that during concurrent training, the exercise modality (running vs. cycling) during the sprint interval exercise (when performed first) has no effect on mechanical output during the subsequent resistance exercise session but has an influence on subjective ratings."

Ref 26 and 33 have no doi.

Response: The DOI for reference 26 has been added. Reference 33 don't have any DOI.

Round 2

Reviewer 1 Report

Dear authors

Thank you for your efforts and for the consideration of our suggestions

Sincerely

Reviewer 2 Report

Please ensure consistency throughout the paper when referring to the exercise modality as “sprint interval” is used in the title but “high-intensity” in the abstract. Check the manuscript for consistency.